# TBI and Tau Loss of Function Both Affect Naïve Ethanol Sensitivity in *Drosophila*

**DOI:** 10.3390/ijms25063301

**Published:** 2024-03-14

**Authors:** Valbona Hoxha, Gaurav Shrestha, Nayab Baloch, Sara Collevechio, Raegan Laszczyk, Gregg Roman

**Affiliations:** 1Department of Biology, Lebanon Valley College, Annville, PA 17003, USA; nayab.baloch@icahn.mssm.edu (N.B.); sc012@lvc.edu (S.C.); rrl001@lvc.edu (R.L.); 2Department of Biology, University of Mississippi, Oxford, MS 38677, USA; gshrest2@go.olemiss.edu; 3Department of BioMolecular Sciences, University of Mississippi, Oxford, MS 38677, USA; groman@olemiss.edu

**Keywords:** TBI, alcohol, *Drosophila*, *tau*

## Abstract

Traumatic brain injury (TBI) is associated with alcohol abuse and higher ethanol sensitivity later in life. Currently, it is poorly understood how ethanol sensitivity changes with time after TBI and whether there are sex-dependent differences in the relationship between TBI and ethanol sensitivity. This study uses the fruit fly *Drosophila melanogaster* to investigate how TBI affects alcohol sensitivity and whether the effects are sex-specific. Our results indicate that flies have a significantly higher sensitivity to the intoxicating levels of ethanol during the acute phase post-TBI, regardless of sex. The increased ethanol sensitivity decreases as time progresses; however, females take longer than males to recover from the heightened ethanol sensitivity. Dietary restriction does not improve the negative effects of alcohol post-TBI. We found that *tau* mutant flies exhibit a similar ethanol sensitivity to TBI flies. However, TBI increased the ethanol sensitivity of *dtau^KO^* mutants, suggesting that TBI and *dtau* loss of function have additive effects on ethanol sensitivity.

## 1. Introduction

Traumatic brain injuries (TBIs) are one of the leading causes of neurological disabilities and death worldwide [1,2,3]. TBI-associated disabilities may cause a broad range of short- and long-term neurological, physiological, motor, and behavioral impairments [4,5,6]. TBI is prevalent among young adults, even more so for those involved in recreational and competitive sports [7,8,9]. TBI is also common among US veterans who have served in Afghanistan and Iraq [10,11].

TBI outcomes are divided into primary and secondary damage. Primary damage occurs during the initial mechanical impact to the brain, resulting in brain tissue damage, axonal damage, edema or blood–brain barrier impairment, and intestinal barrier dysfunction [12]. Secondary injuries include cellular and molecular changes that occur over time due to the primary injuries. Secondary damage to the brain is initiated by damaged cells that release various intracellular proteins in the extracellular space, leading to the activation of cytokines and chemokines (e.g., IL-1α, IL-6, IL-10, interferon-gamma (IFN-γ), and monocyte chemoattractant protein-1), and reactive oxygen species (ROS) [13]. Together with these immune responses, brain injury also leads to activation of the intrinsic apoptotic pathway through the activation of sFas and Caspase 3 [14,15].

Previous work in human and mouse models has shown that TBIs are linked to an increased risk of alcohol self-administration and alcohol use disorders (AUDs) [7,16,17,18]. In humans, TBI effects on alcohol use/misuse have been found prominently in young adults engaged in sports or military veterans [7,16,17]. Furthermore, some studies indicate that TBI is associated with an increase in the frequency of binge drinking [19,20] and higher sensitivity to the sedative effects of ethanol post-TBI [16,18].

Understanding the mechanisms responsible for how TBIs influence the probability of AUDs is likely essential for the development of effective pharmacological interventions; however, these mechanisms remain largely unknown. Some indications for the connecting mechanisms may come from the time courses of disease development after TBI and the sex differences in the emergence of AUDs after TBIs. Research on the effects of alcohol-related behaviors in humans inflicted with TBI often considers the acute phase post-brain injury; however, the time needed for the brain to recover following brain injury is currently understudied. Furthermore, most of the studies do not account for sex differences, focusing more on males [4,7,16,18], since males account for the majority of TBIs in the US [21,22], and male drinkers tend to drink more heavily and more frequently than females [23]. While the effects of alcohol post-TBI are understudied, there is a large body of literature looking at the role of sex after TBI alone. However, there are significant sex differences in the responses to TBIs in patients and animal models [24,25,26,27]. There are also substantial sex differences in the escalation of self-administration and brain damage associated with excessive alcohol consumption [28,29,30,31,32]. Hence, the time course of ethanol responses following TBI and sex differences should provide valuable information as to the biological changes that account for differences in AUD susceptibility following TBI. Additional insights on the molecular and cellular mechanisms connecting TBI and AUDs would likely come from the development of new animal models for TBI-induced changes in ethanol-related behavior and neurobiology.

*Drosophila melanogaster* is an important model organism for studying ethanol-related behaviors and TBI due to the behavioral and molecular mechanisms that are similar to those of mammals and the accessible and powerful genetics available for this species [33,34,35,36]. In this study, we used *D. melanogaster* to evaluate the effect that time post-TBI and sex have on ethanol sensitivity. Adult flies were exposed to a single high dose of 70% ethanol at different time intervals after TBI, and the sedation time of the flies was measured. Fly mortality rates were determined 24 h after ethanol exposure. In addition, we investigated if dietary restriction affects ethanol sensitivity post-TBI [35].

At the molecular level, one of the major hallmarks of TBI is the accumulation of aberrant Tau protein in the form of Tau hyperphosphorylation or aggregates generating neuro-fibrillary tangles (NFTs) [37,38,39,40]. Tau is expressed predominantly in neurons, although it may also be expressed in glia [41]. In neurons, Tau is either bound to the inner side of the plasma membrane or to microtubules, where it stabilizes the microtubules or assists in motor-driven axonal transport [42,43]. In addition, Tau can also bind dendritic f-actin to help stabilize the cytoskeletal elements in spines [44]. Mutations in the human Tau gene, cause frontotemporal dementia and neurodegeneration, leading to diseases such as Parkinson’s or Alzheimer’s. Tau pathology is also observed in all forms of TBI, ranging from mild to severe, and strongly correlates to TBI severity [45,46].

*Drosophila* Tau (dTau) is 44% identical and 66% similar to human Tau, and is predominantly expressed in the larval brain, ventral cord (VNC), and peripheral nervous system (PNS). In adult *Drosophila*, Tau expression is mainly located in the imaginal discs and adult retina and brain, including the mushroom bodies [47]. dTau is a true vertebrate ortholog, and its loss alters brain cytoskeletal dynamics and affects neuronal plasticity, causing retinal degeneration and impaired habituation [48,49]. To elucidate the importance of dTau in ethanol sensitivity post-TBI, we used the *dtau^KO^* mutants and flies expressing *dtau* RNAi, and assessed their role in alcohol sensitivity, with and without TBI.

## 2. Results

### 2.1. TBI Paradigm

It has previously been shown that the number of head strikes a fly receives with the HIT device increases the mortality rate [34]. We also found that increasing the number of strikes increases the mortality rate 24 h after TBI (MI24) in both males and females (Figure 1). The flies exhibit a mild mortality rate after two and four strikes, with a 4–13% mortality rate and a moderate to severe mortality rate as the number of strikes increased (6–10 hits). Furthermore, our results indicate that the effect of TBI on mortality is found in both sexes, inflicting a similar mortality rate on both males and females (Figure 1). These data are consistent with prior findings [34]. Hence, our assay is reproducible and consistent. Our data further suggest that four strikes produce a relatively mild TBI, with 90–95% of the flies surviving and having apparent normal mobility 10–15 min after the strikes. We, therefore, used four strikes separated by five-minute intervals as our standard protocol for all subsequent experiments.

### 2.2. TBI Alters Ethanol Sensitivity

An increasing body of evidence indicates that TBIs are a risk factor for AUDs [50]. To determine if TBIs change the behavioral responsiveness to ethanol in *Drosophila*, we initially tested the sedation sensitivity of males and females exposed to a high ethanol concentration (70%). Flies were placed in 70% ethanol, and the time required for 50% of the flies to sedate (ST50) was recorded. Ethanol sensitivity was measured 2, 24, and 48 h after TBI (Figure 2). Our results indicate that males were very sensitive to the effects of ethanol during the acute TBI period (2 h later). However, ethanol sensitivity improved as time passed (24 h later) and was restored to normal after 48 h. Surprisingly, we found that control flies were also slightly more sensitive to ethanol 2 h later compared to 24 and 48 h later, indicating that stress also plays an important role in ethanol sensitivity. Nevertheless, alcohol sensitivity in flies with TBI was significantly lower than in the control during the acute TBI period and was restored to normal levels after 48 h (Figure 2a).

Prior studies of the role of sex on human brain injury are contradictory, with some studies finding no sex differences [34], while other studies indicate that females are more likely to die from TBI than males [51,52]. Sexual dimorphism in alcohol-related behaviors is also found in many organisms, including *Drosophila* [30,53]. Thus, we also tested the effect of ethanol sensitivity post-TBI in females (Figure 2b). Our results indicate that during the acute phase post-TBI, both males and females are more sensitive to the sedative effects of ethanol (*p* < 0.01 and *p* < 0.0001, respectively). Surprisingly, while, during the acute phase post-TBI, there is no sex difference in ethanol sensitivity, females remain sensitive to the effects of ethanol for a longer period compared to males (Figure 2c). These results demonstrate that TBI strongly affects ethanol sensitivity during the acute phase post-TBI. However, as time progresses, ethanol sensitivity recovery is faster in males than females.

### 2.3. Ethanol Exposure Post-TBI Increases the Mortality Rate

Next, we examined whether the high dose (70%) of ethanol given during the ethanol sensitivity assay affects the mortality rate. Flies exposed to alcohol post-TBI were transferred to a vial with food and water access and allowed to rest. The mortality rate (MI24) was recorded 24 h later. The mortality rate of male flies exposed to ethanol 2 h post-TBI was 37.9 ± 7.1%, which is significantly higher than control TBI-treated flies that were not exposed to ethanol (Figure 3). When the flies were exposed to ethanol 24 and 48 h after TBI, the mortality rate was not significantly different from the control, unexposed flies, demonstrating a time-limited effect of TBI on ethanol-induced mortality (Figure 3).

### 2.4. Post-TBI Diet Does Not Affect Ethanol Sensitivity and Mortality

Previous work in rodents and *Drosophila* indicates that diet may play an influential role during the acute phase of recovery from traumatic brain injury [35,54,55,56]. In *Drosophila*, a water-only (starvation) diet for up to 24 h post-TBI significantly improves the survivorship of flies compared to a food diet [35]. Hence, we examined whether diet post-TBI impacts the sensitivity to the sedative effects of ethanol. Male and female flies were placed on food or a starvation diet following injury for 24 h (or until an ethanol sensitivity assay was performed; Figure 4). This dietary manipulation immediately after the TBI does not affect the sedative effects of ethanol during the acute phase (2 h) post-TBI (Figure 4a,b). Diet post-TBI also does not influence ethanol sensitivity 24 h after the TBI (Figure 4a). Interestingly, as seen before (Figure 3), females are slower to recover normal sensitivity from the sedation effects of ethanol 24 h post-TBI, and diet does not alter this sensitivity (Figure 4b). Our results indicate that, while diet is important for attenuating some TBI-associated symptoms, it does not play a significant role in acute ethanol sensitivity post-TBI.

Since an intoxicating ethanol exposure post-TBI affects mortality, especially during the acute phase post-TBI, we further examined if diet post-TBI plays a role in ethanol-induced mortality (Figure 4c). Mortality rates were measured 24 h after acute ethanol exposure following brain injury. Consistent with the ethanol sedation sensitivity data, diet did not improve mortality rates (MI24). Our results suggest that, while a starvation diet can strongly influence post-TBI recovery [35], it does not affect ethanol sensitivity post-TBI in *Drosophila*.

### 2.5. Loss of dTau Exacerbates Acute Ethanol Sensitivity

In vertebrates, brain injury leads to axonal injury and accelerated Tau pathology [37]. To determine whether *dtau* affects responses to ethanol sensitivity following TBI in *Drosophila*, we examined the ethanol sensitivity of *tau* mutants with and without TBI (Figure 5). We used a knock-out (*tau^KO^*) mutant lacking exons 2–6, including the tubulin-binding repeats; this deletion results in a null mutation lacking the 50 kDa and 75 kDa dTau isoforms [48]. The *tau^KO^* mutants exposed to acute ethanol sensitivity exhibit increased sensitivity to acute ethanol compared to wild-type Canton-S (CS) flies (Figure 5), regardless of sex (*tau^KO^* male vs. CS male *p* < 0.0001 and *tau^KO^* female vs. CS female *p* < 0.0001).

In this experiment, we also subjected male and female wild-type CS and *dtau* mutants to TBI, and measured changes in acute ethanol sensitivity 2 h post-TBI. Heterozygous *dtau^KO^* male flies showed modest ethanol sensitivity compared to the control flies; *dtau^KO^* homozygous flies showed a significant decrease in ethanol sensitivity (*p* < 0.0089). Homozygous *dtau^KO^* mutants, regardless of sex, show levels of ethanol sensitivity similar to CS-TBI (*p* > 0.05). Consequentially, TBI increased the ethanol sensitivity of the *dtau^KO^* homozygotes, indicating that the chronic effect of TBI is at least partially independent on *dtau*.

To further validate our observation that Tau expression is important for ethanol-related behaviors, we measured rapid ethanol tolerance using the loss of righting reflex (LORR) [57]. This LORR assay was used on *dtau^KO^* mutants and UAS-*dtau* RNAi transgenes driven exclusively in the nervous system by nSyb-Gal4 (Figure 6). Since our previous experiments suggested that the phenotype of *dtau^KO^* is not sex-specific, we used only males for these experiments. The results of the LORR assay are consistent with those obtained by the previous ethanol sensitivity assay, showing that *dtau^KO^* mutants exhibited an increased ethanol sedation sensitivity as indicated by lower LORR (Figure 6a). Therefore, using two different assays, we found that homozygous *dtau^KO^* null mutant males are sensitive to both ethanol sensitivity and rapid tolerance. Interestingly, the *dtau^KO^* mutants also displayed a reduced rapid ethanol tolerance, suggesting that dTau function may also be important for the neuroplasticity underlying tolerance formation (Figure 6a).

Next, we wanted to validate the effect of *dtau* using the targeted expression of *dtau* RNAi in the nervous system using the pan-neuronal driver *nSyb*-Gal4 (Figure 6b). In this experiment, the *nSyb*-Gal4 and the *dtau* RNAi line on their own were significantly different from each other, suggesting that one of the two elements may have a small effect on ethanol sensitivity. Nevertheless, similar to the *dtau^KO^* mutants, flies with reduced *dtau* activity due to the expression of *dtau* RNAi in neurons exhibited increased ethanol sedation sensitivity compared to both background control lines. Hence, the *dtau* gene is critical in mediating the effects of ethanol sensitivity and rapid tolerance.

To observe if the increased ethanol sensitivity found after TBI or in the *dtau* loss-of-function mutant flies was caused by altered ethanol pharmacokinetics, we measured the internal alcohol absorption of wild-type Canton-S flies in the presence or absence of TBI, and in the *dtau^KO^* mutants. The tissue alcohol levels in flies exposed to TBI were not significantly different from the treatment controls (Figure 7a). Moreover, the ethanol absorption in the *dtau^KO^* mutants was also not significantly different from Canton-S (Figure 7b). These data support the hypothesis that TBI and *dtau* loss of function alter ethanol sedation sensitivity through functional changes in neuronal activity and not through differences in alcohol tissue concentrations.

## 3. Discussion

Herein, we have shown that TBI in *Drosophila* increases naïve ethanol sedation sensitivity. This increase in sedation sensitivity lasts for at least 24 h post-TBI, with some effects still seen in females 48 h after TBI treatments. The differences in female ethanol sensitivity could be due to females having eggs and more fat present (higher body mass) to either buffer or add to TBI severity and EtOH remaining for longer periods due to being absorbed into fat. However, prior studies on humans indicate that females are more sensitive to the levels of alcohol compared to males [58]. In humans, overall, females are more vulnerable than males to many of the health consequences of alcohol use, suggesting that TBI could also play a role in the increased alcohol sensitivity observed in female flies.

The effect of TBI on ethanol sensitivity is seen in both males and females and is unaffected by starvation following TBI. Moreover, ethanol exposure 2 h after TBI significantly increases mortality compared to TBI alone. However, this effect was gone by 24 h post-TBI, suggesting acute TBI further increases ethanol intoxication sensitivity. We have also seen an increased ethanol sedation sensitivity in the Oregon R and Canton-S genetic background strains. Hence, the ability of TBIs to increase the sensitivity to ethanol is robust and largely reversible over time.

We have further shown that loss-of-function *dtau* mutants have an increased ethanol sedation sensitivity phenotype, mimicking the effect of TBI. To further elucidate if the ethanol sensitivity phenotype is mainly because of *dtau^KO^* expression in neurons, and not as a result of glial cells in the brain, we used the pan-neuronal knockdown of *dtau* with RNAi. The expression of *dtau*-RNAi exclusively in the neurons also leads to an increase in ethanol sensitivity. This result verifies that reducing *dtau* function increases ethanol sensitivity and shows that this requirement for *dtau* is neuronal. Nevertheless, the effect of TBI on ethanol sedation sensitivity is additive with the *dtau^KO^* mutation. This additivity strongly suggests that the underlying causes of increased ethanol sedation sensitivity in TBI and *dtau^KO^* mutants are different.

### 3.1. TBI and Ethanol Sensitivity

TBIs in *Drosophila* are responsible for several types of injury, several of which may be at least partially responsible for an increase in ethanol sedation sensitivity and ethanol-induced mortality [59,60], including, but probably not limited to, disruption of the blood–brain barrier [34,61], mitochondrial dysfunction [62,63], and acute neuroinflammation [34,35,36,64]. The disruption of the *Drosophila* blood–brain barrier function can lead to changes in naïve ethanol sensitivity and tolerance formation [65,66,67]. An insertional mutation of the *moody* GPCRs, which are preferentially expressed in the subperineural glia, results in a decreased barrier function and increased resistance to ethanol sedation [66]. The disruption of blood–brain barrier function by TBI may also be expected to decrease ethanol sensitivity rather than increase sensitivity, as we have seen, assuming the acute response after injury is the same as the chronic disruption seen in *moody* mutants. However, the subperineural glia knockdown of the *scarface* serine protease results in increased ethanol sedation sensitivity without affecting the paracellular permeability of the blood–brain barrier [67], suggesting there may be multiple ways for perturbing the blood–brain barrier function that affect ethanol sensitivity. Further work is required to determine if the *moody* or *scarface* pathways functionally interact with acute TBI to modify naïve ethanol sensitivity.

TBIs may also affect ethanol sensitivity through changes in mitochondrial function, although the effects of mitochondrial function on naïve alcohol sensitivity remain relatively unexplored [68]. In *Drosophila*, proteins required for the transport of mitochondria down axons are also required in Dopaminergic neurons to form a positive association between the hedonistic effects of ethanol and an odor [69]. This requirement is specific to the rewarding properties of ethanol since the sugar reward was unaffected. In rodents, a single intoxicating dose of alcohol impacts mitochondrial function [70]. In *C. elegans*, intoxicating doses of ethanol lead to mitochondrial fragmentation [71]. In Zebrafish, the acute presentation of ethanol stimulates mitochondrial respiration and O_2_ consumption [72]. Hence, across species, there appear to be exposure-induced differences in how mitochondria respond to alcohol, with changes in transport and function. Acute TBI-induced changes in mitochondrial function may sensitize neurons to ethanol inhibition, leading to increased sedation sensitivity.

TBI in *Drosophila* may also increase ethanol sedation sensitivity by activating neuroinflammation pathways [73,74]. TBIs activate the two main branches of the innate immune system: the immune deficiency pathway (IMD) and the toll pathway [34,35,36,59]. Sedating ethanol concentrations increases the transcription of genes within the toll pathway [75,76]. Sedating doses of ethanol also activate the toll pathway [74]. Genetically activating the toll pathway increases ethanol resistance, while inhibiting the pathway increases ethanol sensitivity [74]. Since TBIs also activate toll signaling, this injury may naïvely be expected to drive an increased resistance rather than increased sensitivity, as we have found. This assumption, however, does not account for differences in where toll signaling is employed and the outputs prioritized in the injured tissue. More information is needed on the specific effects of TBI on the toll pathway before we understand how TBI-induced neuroinflammation affects ethanol sedation sensitivity. It is not currently known if the TBI-activated IMD pathway also affects ethanol sensitivity.

The Janus kinase/signal transducer and activator of transcription (JAK/STAT) and c-Jun N-terminal kinase (JNK) pathways also potentially link TBI-induced immune responses and changes in alcohol responsiveness. In *Drosophila*, TBI induces both the JAK/STAT and JNK pathways [64,77]. These two pathways are also activated by axonal injury and interact to regulate axon regeneration [78,79]. Interestingly, the knockdown of STAT92E, the only STAT ortholog in *Drosophila*, led to significantly higher ethanol-induced locomotor behavior in flies previously exposed to ethanol [80]. It is not currently known if this increased sensitivity to the activating properties of ethanol translates to an increased sensitivity to the sedating effects of this drug [80]. It remains possible that the loss-of-STAT92E activity would impair post-TBI axon recovery, leading to an increased sensitivity to ethanol sedation.

TBIs in *Drosophila* lead to wide-ranging tissue damage and the activation of stress and injury responses [59]. In addition to the pathways listed above, there are likely other effects of these head injuries that may also acutely influence the ethanol sensitivity we have found after the HIT protocol (e.g., calcium dysregulation [81]). It will be interesting to see how activating or inhibiting each of these known responses to TBI will affect acute ethanol sedation sensitivity and mortality. Since these pathways exhibit differences in when they are activated and required for recovery from TBIs, it will also be meaningful to see how these individual pathways alter the time course of TBI-induced hypersensitivity to ethanol.

### 3.2. Tau and Ethanol Sensitivity

Analogous to its vertebrate ortholog [82,83], dtau stabilizes microtubules, promotes actin polymerization and cytoskeletal dynamics, and is a negative regulator of translation; dtau has physiological roles in dendrites, nuclei, and axons [48]. In humans and vertebrate animal models, mild or severe TBIs are associated with Tau pathology post-injury [37,84]. Most effects of Tau on axonal degeneration are believed to be due to toxic gain-of-function posttranslational modifications of Tau and not loss-of-function ones [41,85,86]. The exact role of *dtau* in *Drosophila* TBI-induced disorders is not currently known. Similar to mouse *tau* knockout mutants, *dtau^KO^* flies have a normal survival rate and fertility [87,88]. Moreover, the *dtau^KO^* mutants maintain sexually dimorphic responses to TBI, suggesting that *dtau* does not play a role in these sex-dependent differences in TBI outcomes in *Drosophila* [89].

Several genes required for learning and memory in *Drosophila* also have ethanol sensitivity defects, pointing to shared synaptic mechanisms [90,91,92]. Loss-of-function *dtau^KO^* mutants have defects in habituation to electric foot shocks and demonstrate an increased ability to form protein-synthesis-dependent long-term memories (PSD-LTM) [48]. The reduction of Tau expression through the expression of RNAi transgenes in the α’/β’ mushroom body neurons was sufficient to generate both habituation of PSD-LTM phenotypes [48]. Future work will establish if dTau’s requirement for normal ethanol sensitivity will be similarly localized to the α’/β’ mushroom body neurons.

## 4. Materials and Methods

### 4.1. Fly Stocks and Maintenance

Oregon R flies were used as the control strain in the experiments shown in Figure 1, Figure 2, Figure 3 and Figure 4. Assays were performed on 4- to 7-day-old males or females that were not selected as virgins. The dtau^KO^ (RRID:BDSC_64782) flies were a gift from Dr. E.M. Skoulakis (Institute for Fundamental Biomedical Research, Vari, Greece). Flies were reared on instant *Drosophila* medium (Carolina Biologicals, Burlington, NC, USA) at 70% relative humidity.

Canton-S (CS) wild-type flies were used for the experiments shown in Figure 5, Figure 6 and Figure 7. These flies were raised on Cornmeal food at 25 °C in a 12:12 h light–dark cycle as previously described [93]. The *dtau^KO^* mutation and the *nSyb*-Gal4 driver (RRID:BDSC_51635) were backcrossed into the resident Cantonised-*w*^1118^ control isogenic background for six generations for this experiment. The UAS-*dtau*RNAi (RRID:BDSC_40875; P{*y*[+t7.7] *v*[+t1.8] = TRiP.HMS02042}attP40; gift from Dr. E.M. Skoulakis) was initially outcrossed for six generations into a Cantonized-*y*^1^ background, before replacing the X chromosome with a wild-type Canton-S X chromosome. These flies were raised on a Standard Cornmeal Medium [94].

### 4.2. Traumatic Brain Injury (TBI) Paradigm

For these experiments, mature flies 4–7 days old were used. Flies were separated by sex, and 10–12 flies were placed into a *Drosophila* food vial. A TBI apparatus (HIT device) was constructed by connecting a tension spring to a board as previously described [34]. A 15 mL conical tube was placed in a fixed position inside the spring to stabilize the fly vial to the concussion chamber spring. The narrow vials with flies could fit inside the conical tube, creating a stable and reproducible impact between the fly vial and the concussion chamber (Figure 6). The string was deflected at a 90° angle and released on a pad. The velocity of the impact was ~3.0 m/s (6.7 miles/h).

To reduce the impact of the hit, half a plug was placed on the bottom of the vial. The flies were confined to the bottom 1 cm of the vial and let to rest for 5 min. The vial was then connected to the free end of the spring. The spring was deflected at a 90° angle and released on the pad. There was a 5 min recovery period between each strike. Following the completion of the injury, all fly cohorts were put into vials, which were placed on their sides to allow for a full recovery. Control flies were placed under the same conditions for the duration of the concussions; however, they were not subjected to any strike. Each experiment was reproduced at least eight times (n = 80–130 flies). While the mortality rate was <10%, dead flies were excluded from further experiments. All original data can be accessed in the Appendix A.

### 4.3. Ethanol Sensitivity Assay and Mortality Rate

All the flies were separated into males and females. The ethanol sensitivity assay was as previously described [95] with slight modifications. In short, the treatment groups and the control were placed in an exposure chamber (vial) by placing half of a plug at the bottom of the chamber. The chambers had the same air space, allowing an equal ethanol concentration to volume ratio. Next, 8–12 flies were placed into the ethanol exposure chamber and left to rest for 4 min. Then, 1000 µL of 70% pure ethanol was added to a plug. Flies were subjected to the vial plug with ethanol, and the rate of ethanol sedation in each group was recorded every 4 min for a maximum of 60 min. ST50 was calculated as the median sedation time determined when half of the flies first became stationary (they were intoxicated and, therefore, lost balance and fell). Mortality rate 24 h (MI24) after ethanol exposure was also measured and recorded.

In Figure 6, the *dtau^KO^* and *dtau*-RNAi mutants and transgenic lines were examined using the loss of righting reflex (LORR) ethanol sensitivity assay [57]. In this assay, 50% ethanol vapor is generated through a constant airflow of about 500 mL per minute and distributed to different vials containing 30 flies each. The time taken for 50% of flies to become sedated is measured by observing the loss of the righting reflex (LORR) [57]. Naïve flies are exposed to 50%/50% ethanol/water vapor for ethanol sensitivity, and time for 50% LORR is recorded. For rapid tolerance, 90% of naïve flies are sedated with 50%/50% ethanol/water vapor and incubated at room temperature for 4 h for recovery, and further tested with a second exposure of 50% ethanol for the measurement of the time taken for 50% LORR of flies to be sedated. Rapid tolerance is calculated by subtracting 50% LORR of 1st exposure from 2nd exposure [96]. Alcohol absorption was measured as previously described [93] using a colorimetric kit (Abcam, Cambridge, UK; ab65343). All original data can be accessed in the Appendix A.

### 4.4. Diet Studies

Flies were separated into males and females. Each sex was placed in food or starvation (water-only) diet for a maximum of 24 h (or until ethanol sensitivity assay was performed). Following the treatment, flies were placed in an ethanol exposure chamber and ethanol sensitivity was measured. Control flies were placed into different vials similar to TBI-exposed flies, to control for the effect of handling and stress, but were not subjected to any strikes. Flies were later placed in food or water-only diet similar to the TBI flies until ethanol sensitivity assay was performed. All original data can be accessed in the Appendix A.

### 4.5. Statistical Analyses

Statistical analyses were performed using GraphPad Prism, version 8.4.2 and version 9, and by SigmaPlot, version 15. Figures were made using GraphPad Prism, version 8.4.2. Data failing Shapiro–Wilk normality tests were log10-transformed before ANOVA [97]. Statistical tests are specified in the figure legends. Experimental and control flies were always tested simultaneously. Following initial ANOVA, the Tukey multiple comparison test was performed. All data are expressed as means ± SE. The Kruskal–Wallis test was used to evaluate differences in the formation of rapid tolerance between Canton-S and *dtau^KO^* mutants [98].

## Figures and Tables

**Figure 1 ijms-25-03301-f001:**
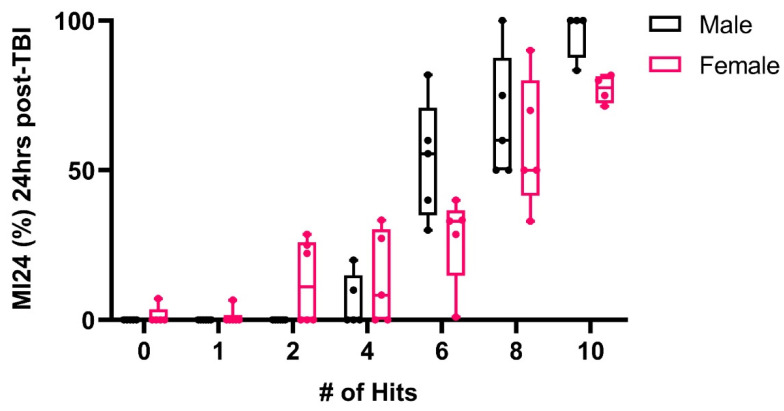
Development of the TBI model. Flies were subjected to different numbers of strikes, and the percentage of mortality rate 24 h post-concussion (MI24 %) was recorded (n = 5–6 groups).

**Figure 2 ijms-25-03301-f002:**
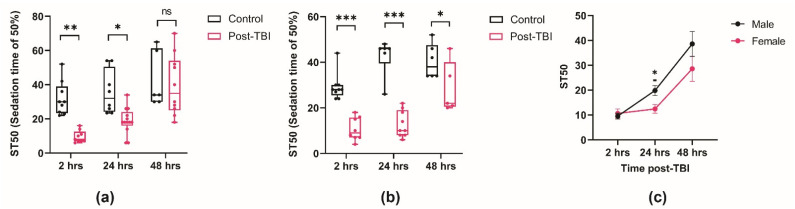
Male and female flies were subjected to TBI, and ethanol sensitivity was measured at 2 h, 24 h, and 48 h post-TBI. (**a**) Male flies exposed to ethanol post-TBI, and then compared to control flies exposed to ethanol alone (two-way ANOVA: F(2,49) = 12.34; ** *p* < 0.01, * *p* < 0.05, ns = not significantly different; n = 5–12 groups). (**b**) Female flies exposed to ethanol post-TBI, compared to control flies exposed to ethanol alone (two-way ANOVA: F(2,37) = 14.27; *** *p* < 0.001, * *p* < 0.05; n = 5–9 groups). (**c**) The effect of ethanol, administered at different times post-TBI, is shown. Males were compared to females at each timepoint by two-way ANOVA, followed by Tukey’s multiple comparison test (F(2,47) = 26.94; * *p* < 0.05; n > 6 groups).

**Figure 3 ijms-25-03301-f003:**
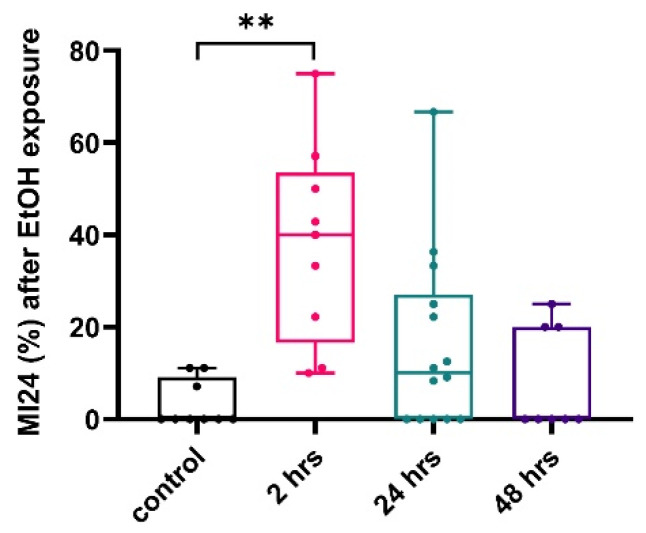
TBI increases ethanol-induced mortality. % mortality post ethanol exposure in TBI-induced male flies. Control flies were exposed to ethanol alone and MI24 was assessed 24 h later. The other flies were exposed to ethanol 2 h, 24 h, or 48 h post-TBI. MI24 (%) was assessed 24 h after ethanol exposure ** *p* = 0.002, n > 8 groups, Kruskal–Wallis.

**Figure 4 ijms-25-03301-f004:**
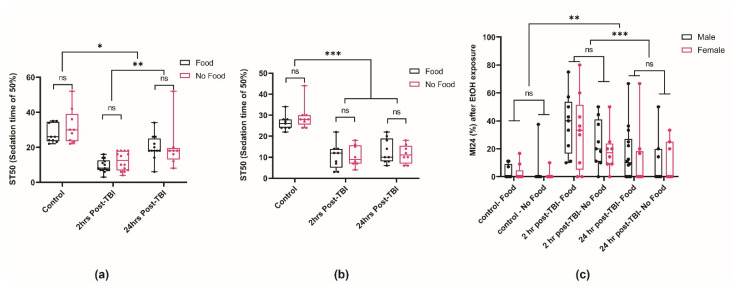
Starvation after TBI does not affect ethanol sedation sensitivity. (**a**) Ethanol sensitivity was measured in control male flies without TBI, 2 h post-TBI, or 24 h post-TBI. Kruskal–Wallis two-way ANOVA on ranks, factor; food, *p* = 0.67, time post-TBI * *p* < 0.011, ** *p* < 0.002, ns = not significantly different, n > 8 groups. (**b**) Ethanol sensitivity was measured in control female flies without TBI, 2 h post-TBI, or 24 h post-TBI. Kruskal–Wallis two-way ANOVA on ranks, factor; food, *p* = 0.741, time post-TBI *** *p* < 0.001, ns = not significantly different, n > 8 groups. (**c**) The mortality 24 h after ethanol exposure was measured for male and female flies that received a TBI at the indicated times with the indicated post-TBI diet. (three-way ANOVA followed by Tukey’s multiple comparisons test: Comparison factor: time post-TBI, ** *p* < 0.014, *** *p* < 0.001; diet and sex, *p* > 0.05, ns = not significantly different, n > 7 groups).

**Figure 5 ijms-25-03301-f005:**
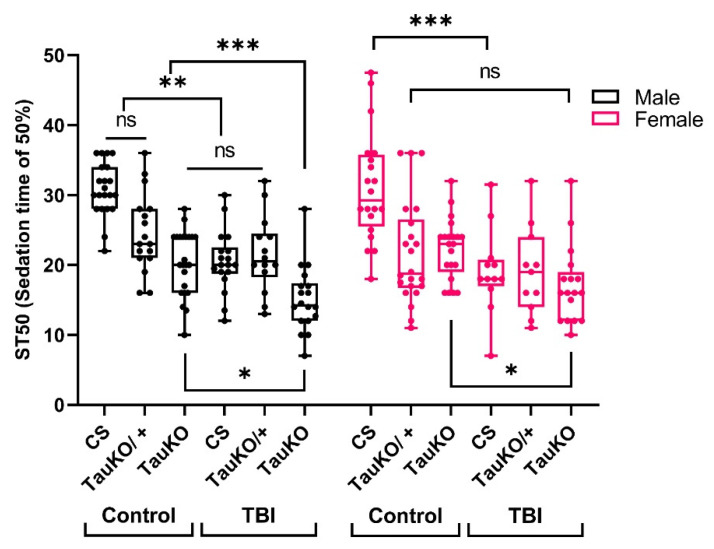
TBI and *dtau* loss of function produces an additive effect in ethanol sedation sensitivity. *Dtau^KO^* mutants and heterozygous were subjected to TBI and tested for ethanol sedation sensitivity with Canton-S (CS) as the wild-type control. (Three-way ANOVA: Tukey’s multiple comparison test, n > 11 groups, * *p* < 0.0450, ** *p* < 0.0089, *** *p* < 0.0003, ns = not significantly different).

**Figure 6 ijms-25-03301-f006:**
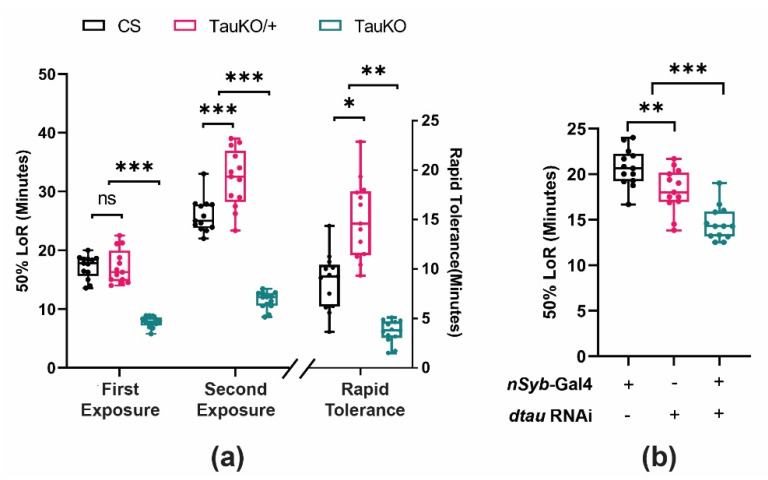
Loss-of-function *dtau* homozygous mutants exhibit increased ethanol sedation sensitivity and altered rapid tolerance. (**a**) Males that were either heterozygous or homozygous for the *dtau^KO^* mutation were examined for loss of righting reflex in the FlyBar assay [57]. Wild-type Canton-S (CS) was used as a positive control. (Two-way ANOVA, Tukey’s test for 1st and 2nd exposure; comparison for factor: treatment, *** *p* < 0.001, ns = not significantly different; genotype within 1st and 2nd exposure, *** *p* < 0.001, n = 13 groups. Kruskal–Wallis one-way ANOVA on ranks for rapid tolerance, * *p* < 0.027, ** *p* < 0.01, n = 13 groups). (**b**) Flies expressing *dtau* RNAi transgene in the brain (through nSyb-Gal4) were subjected to ethanol sedation sensitivity (one-way ANOVA, Tukey’s test; ** *p* < 0.012; *** *p* < 0.001, n = 13 groups).

**Figure 7 ijms-25-03301-f007:**
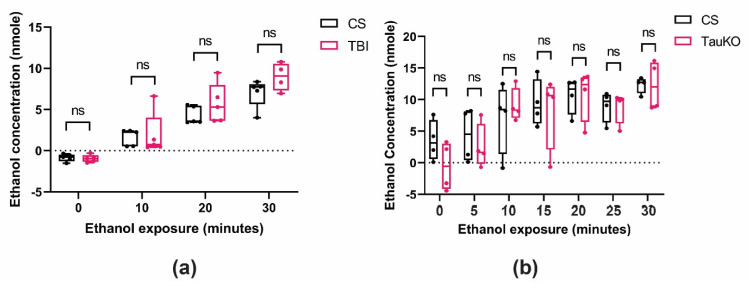
Neither acute TBI nor *dtau* loss of function alters ethanol absorption in *Drosophila*. (**a**) To test for ethanol absorption, wild-type flies were tested in the presence and absence of TBI. Flies were subjected to ethanol at 10 min intervals (two-way ANOVA, Tukey’s test, comparisons for factor: treatment; *p* = 0.099, time; *p* < 0.005, n = 4–5 groups). (**b**) Wild-type CS and *dtau^KO^* flies were tested for ethanol absorption at 5 min intervals (n = 4 groups, two-way ANOVA, Tukey’s test, comparison factor: genotype; *p* = 0.466, time; *p* < 0.001), ns = not significantly different.

## Data Availability

The data presented within the article and Appendix A is openly available in eGROVE at https://egrove.olemiss.edu/pharmacy_facpubs/287/.

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
