# Peer review of "TBI and Tau Loss of Function Both Affect Naïve Ethanol Sensitivity in Drosophila"

_ijms, 2024, doi:10.3390/ijms25063301_

Round 1

Reviewer 1 Report

Comments and Suggestions for Authors

Title: TBI and Tau loss-of-function both affect naïve ethanol sensitivity in Drosophila

The manuscript is basic and may not offer any more insights for the readers of the IJMS. Insufficient quantities of tested flies were also seen, despite the simplicity of the experiments. This causes this reviewer concern over the data representativeness.

1.       The similarity index (exclude refs) = 21%, which is good.

2.       The introduction is inadequate. The authors wrote this section as if they were putting pieces of evidence together without story formation. This further impedes the rationale and knowledge gap of the manuscript.

3.       Line 23: Traumatic brain injury injuries (TBI)? and Drosophila must be italic.

4.       Number of tested flies (n= 5-9) were too low. Since it is simple to produce flies, why did the authors choose such a tiny population? This reviewer would say that this “n” is unacceptable.

5.       70% ethanol is toxic to flies, the authors must provide data of flies without TBI + 70% ethanol as control as well.

6.       Fig6 B. How the authors explain the different between black and pink bar? Should it be the same?

7.       Why did you use two Drosophila media? Any explanation for that?

8.       RT-PCR and/or western blotting must be used to compare the Tau levels of the control vs. TauKO.

9.       One short paper with 97 references is excessive; please reduce it to 50–60 references.

Comments on the Quality of English Language

Minor editing of English language required

Author Response

We thank this reviewer for their careful reading of our manuscript and their thoughtful criticisms.  Our responses in blue text are below each enumerated comment. 

  1. The similarity index (exclude refs) = 21%, which is good.
  2. The introduction is inadequate. The authors wrote this section as if they were putting pieces of evidence together without story formation. This further impedes the rationale and knowledge gap of the manuscript.

We truly appreciate the feedback, and we have made changes to the introduction to further explain the rationale and gap in knowledge.

  1. Line 23: Traumatic brain injury injuries (TBI)? and Drosophilamust be italic.

Thank you for your feedback. We have made the appropriate changes. Line 23 has been changed, and all Drosophila genus have been italicized.

  1. Number of tested flies (n= 5-9) were too low. Since it is simple to produce flies, why did the authors choose such a tiny population? This reviewer would say that this “n” is unacceptable.

Thank you for your insight. In our studies we always use 8-11 flies in every trial, and each trial was repeated 5-9 times (or a population of ~40-90 flies was used). We realize that maybe the description used was not as clear, hence we added the word ‘ groups’ when we refer to the tested flies in the figure descriptions:

        Eg.  (….. n=5-9 groups).

  1. 70% ethanol is toxic to flies, the authors must provide data of flies without TBI + 70% ethanol as control as well.

We truly appreciate this comment, as it is very critical to have a relevant control.

In our experiment we have used the ethanol behavior assay as described by Sandhu et al. 2015 ( reference 93 in our paper). In their paper the authors have used 85% ethanol, without causing toxicity to the flies. Other publications in the field use alcohol concentrations ranging from 65%- 100% (see also doi:10.3791/2541), indicating that the Drosophila they use are still behaving normal after these concentrations. However, in our experiments we tried a range of concentrations and determined 70% ethanol as the lowest concentration required for 50% of the flies to be sedated in 60 minutes of behavioral observation ( STD50%).

We have always used a 70% ethanol as control (without TBI) in all of our experiments. However, after reading the reviewer’s comments we realized that maybe our explanation of Figure 3 might not have been clear enough. We truly appreciate this comment and have clarified Figure 3 to indicate what the control is.

  1. Fig6 B. How the authors explain the different between black and pink bar? Should it be the same?

We have added an additional sentence to the results that addresses this issue (see below).  Although the black and pink bars are relevant background controls, there remains a difference between them. This difference may be caused by the nSybGal4 expression causing increased resistance, or perhaps uncontrolled leaky expression of the dTauRNAi, leading a a slight increase in sensitivity.  Regardless, there is a very large effect when the RNAi is driven pan-neurally.    

“In this experiment, the nSyb-Gal4 and the dtau RNAi line on their own were significantly different from each other, suggesting that one of the two elements may have a small effect on ethanol sensitivity. Nevertheless, similar to the dtauKO mutants, flies with reduced dtau activity due to the expression of dtau RNAi in neurons exhibited increased ethanol sedation sensitivity compared to both background control lines.”

  1. Why did you use two Drosophila media? Any explanation for that?

This paper is a collaboration between Dr. Hoxha and Dr. Roman. The investigators use different Drosophila media to grow their flies, hence there are two different media in the method section.  TBI experiments were performed in Dr. Hoxha’s lab. The Rapid tolerance of Tau flies and ethanol absorption assays were performed in Dr. Roman’s lab.

  1. RT-PCR and/or western blotting must be used to compare the Tau levels of the control vs. TauKO.

We appreciate your comment. The Drosophila TauKO mutant line was generated in the lab of Burnouf et al. (2016) (reference #47). Using RT-PCR and Western blot analysis, the authors showed that the TauKO line expresses no transcript of the Tau gene, nor does it express a protein. We received this same line from Dr. Skoulakis (reference #85) , who replicated the western blot analysis indicating that the line used does not express the Tau protein.

  1. One short paper with 97 references is excessive; please reduce it to 50–60 references.

We appreciate this comment. However, although there is not a lot of data, we think there is a lot of content to address. We believe that the citations used are necessary to provide a complete understanding of the breadth of the work.

Reviewer 2 Report

Comments and Suggestions for Authors

The manuscript “TBI and Tau loss-of-function both affect naïve ethanol sensitivity in Drosophila” by Hoxha et al., presents a good enough English. The authors tested if the traumatic brain injury is associated with alcohol abuse and higher ethanol sensitivity later in life. Furthermore, they investigate how TBI affects alcohol sensitivity and whether the effects are sex-specific.

The manuscript needs major revisions.

Major revision

1)     In material a method. This referee does not believe that the damage to the fly is similar in all of them. The method needs to be better explained.

2)     Authors indicated that flies have a significantly higher sensitivity to the intoxicating levels of ethanol during the acute phase post-TBI. they demonstrated that, but they used no much protocols to demonstrate it.

3)     Why do you think female fly will be more affected respect to male? Authors need to do more experiments with different protocols to demonstrate this. Which differences do you think there are between sex. Differences about Tau, about life span after alcohol consumption, or another one.

4)     Why the authors think TBI and loss of dTau function have additive effects on ethanol sensitivity? They didn't use biochemistry or molecular biology to prove it. Can you do some of those methods? For example, protein extraction and western blot of Tau, mRNA expression of different proteins of interest or other?

5)     The authors indicated that there is an excess of cell death, inflammation and oxidative stress, but they did not measure inflammation or oxidative stress. It would be enough if they measured only some inflammatory and stress-inducing proteins. It would be interesting to know which of the mechanisms could be most altered.

6)     They concluded that neurons will be affected, but what happen with astrocytes and microglia? There are these type of cells in the fly and they intervene in inflammation. They need to discus that or they need to do more experiments about it. This point is so important for this referee.

7)     The study was doing on flies, but author need to add information about human or mice.

8)     Please indicate with type of chemokines and cytokines are produced after alcohol abuse and after TBI damage.

9)     The literature indicates that when NF-kB increases, PGC-1 is phosphorylated and an increase in T-FAM occurs due to the action of NRF1,2. Ultimately, there is an increase in mitochondrial biogenesis. When SIRT-1 is decreased, PGC-1 is not phosphorylated and mitochondrial biogenesis does not occur. Authors should measure mitochondrial biogenesis, using mitotracker to stain mitochondria or immunofluorescence with Tom 20.

Author Response

We are grateful for this reviewer's careful reading and thought comments.   Our responses are in blue below the enumerated comments.  

1) In material a method. This referee does not believe that the damage to the fly is similar in all of them. The method needs to be better explained.

Thank you very much for the feedback. Investigating TBI is indeed hard, and we do agree with the reviewer that using the HIT assay the damage to the flies might not be similar in all of them.

However, the HIT assay has been shown to inflict mechanical injuries that are similar to outcomes characteristic of closed head TBI in humans (Katzenberger et al 2013 and 2016). These outcomes include temporary incapacitation, ataxia, activation of similar innate immune responses, neurodegeneration and death. Furthermore the using the HIT assay, Katzenberger et al. 2013 (https://doi.org/10.1073/pnas.1316895110) showed that risk factors for mortality in flies are shared with humans. Hence, while there might be differences at the single fly level, other studies and our study indicate that the response to the flies at the population level is still significantly different from control flies, suggesting that the HIT assay is a good assay to understand the effects of TBI at the population level.

2)     Authors indicated that flies have a significantly higher sensitivity to the intoxicating levels of ethanol during the acute phase post-TBI. they demonstrated that, but they used no much protocols to demonstrate it.

We appreciate your feedback. To demonstrate the acute effects of ethanol post- TBI, we performed the ‘Ethanol Sensitivity Assay’ that is described in the methods section. What was changed was the time when ethanol was administered to the flies after a TBI (hence the ethanol was administered 2 hrs post-TBI; 24 hrs post-TBI; or 48 hrs post-TBI). We did not write this in the methods sections, to avoid repetitiveness since this is not a separate protocol, but an assessment of the time post-TBI.

3) Why do you think female fly will be more affected respect to male? Authors need to do more experiments with different protocols to demonstrate this. Which differences do you think there are between sex. Differences about Tau, about life span after alcohol consumption, or another one.

We truly appreciate the comment, and we have made some changes to suggest the sex differences in the discussion of the paper.

 "The differences in female ethanol sensitivity could be due to females having eggs and more fat present (higher body mass) to either buffer or add to TBI severity and  EtOH remaining for longer periods of time due to being absorbed into fat. However, prior studies on humans (PMID: 31789554) indicate that females are more sensitive to the levels of alcohol compared to males. In humans, overall females are more vulnerable than males to many of the health consequences of alcohol use, suggesting that TBI could also play a role in the increased alcohol sensitivity observed in female flies."

4)     Why the authors think TBI and loss of dTau function have additive effects on ethanol sensitivity? They didn't use biochemistry or molecular biology to prove it. Can you do some of those methods? For example, protein extraction and western blot of Tau, mRNA expression of different proteins of interest or other?

Thank you for your great suggestions. We have thought about looking more in detail about the causes of these additive effects. At the gene level, we suggest that there might be additive effects. Doing western blot analysis and mRNA expression is beyond the scope of the paper. We appreciate this comment and are looking into these types of experiments in our future work.

5)     The authors indicated that there is an excess of cell death, inflammation and oxidative stress, but they did not measure inflammation or oxidative stress. It would be enough if they measured only some inflammatory and stress-inducing proteins. It would be interesting to know which of the mechanisms could be most altered.

Now that we finished this work, we largely believe that tauopathy is not involved in TBI induced sensitivity. We do wish to look in the future more closely at acute injury mechanisms.

6)     They concluded that neurons will be affected, but what happen with astrocytes and microglia? There are these type of cells in the fly and they intervene in inflammation. They need to discus that or they need to do more experiments about it. This point is so important for this referee.

This is a very important point, and we appreciate you bringing it up. We agree with the reviewer that the TauKO could affect the glial cells. Hence, to verify that the effect we are observing is mainly neuronal, we used a pan-neuronal line (nSyb-Gal4) to express Tau-RNAi exclusively in the neurons. Expression of Tau-RNAi still showed sensitivity to ethanol, suggesting that the effects we are observing are mainly through Tau expression in the neurons.

We think that this comment is very important to all readers, so we have added the suggestions to the paper discussion. Thank you very much for this feedback!

7)     The study was doing on flies, but author need to add information about human or mice.

Thank you very much for your comment. We have addressed this in several places within the introduction.

8)     Please indicate with type of chemokines and cytokines are produced after alcohol abuse and after TBI damage.

We added the information on the different types cytokines and chemokines in the introduction section of the paper.

9)     The literature indicates that when NF-kB increases, PGC-1 is phosphorylated and an increase in T-FAM occurs due to the action of NRF1,2. Ultimately, there is an increase in mitochondrial biogenesis. When SIRT-1 is decreased, PGC-1 is not phosphorylated and mitochondrial biogenesis does not occur. Authors should measure mitochondrial biogenesis, using mitotracker to stain mitochondria or immunofluorescence with Tom 20.

We truly appreciate these suggestions. Thank you very much! We will definitively consider to follow up our studies in our future work to look further look into mitochondrial dynamics.

Reviewer 3 Report

Comments and Suggestions for Authors

A few comments that could be discussed more:

1. Did the authors measure the stress level of animals, and can they do this? At least in mice and rats, pre-experimental or peri-experimental stress is a very strong behavioural regulator (PMID: 10721682, 15325774).

2. Figures are very well presented, but maybe the authors can also make a flowchart describing the experiment's timeline and phases.

3. This paper (PMID: 35389045) also describes the mechanisms against tau pathology, which is possibly worth discussing by the authors.

Author Response

We thank this reviewer for their helpful comments.    Our responses are in blue text below the enumerated comments.  

  1. Did the authors measure the stress level of animals, and can they do this? At least in mice and rats, pre-experimental or peri-experimental stress is a very strong behavioural regulator (PMID: 10721682, 15325774).             Thank you for pointing out the paper by Koks et al. about the relevance of pre-experimental stress and the role of CCK in rats. We have tried to minimize stress in flies, by minimizing handling and letting them rest in the vials that are subjected to TBI or ethanol before doing the assays. Also, when we performed diet studies, we used control flies placed into vials similar to TBI-exposed flies that were not subjected to any strikes.

To account for stress levels, we have modified our results to indicate that stress could also partially account for the results we obtained.

“Surprisingly we found that control flies were also slightly more sensitive to ethanol 2 hours later compared to 24 and 48 hours later, indicating that stress also plays an important role in ethanol sensitivity. Nevertheless, alcohol sensitivity in flies with TBI was significantly lower than in the control during the acute TBI period and was restored to normal levels after 48 hours (Fig 2a).”

  1. Figures are very well presented, but maybe the authors can also make a flowchart describing the experiment's timeline and phases.

The experiments presented in our work are independent from each other. Since different groups of flies were used for the different experiments, it seems unnecessary to place them in a timeline.

3. This paper (PMID: 35389045) also describes the mechanisms against tau pathology, which is possibly worth discussing by the authors.

We have added this valuable reference to our paper.

Reviewer 4 Report

Comments and Suggestions for Authors

Review: TBI and Tau loss-of-function both affect naïve ethanol sensitivity in Drosophila

Mol Sci 2024 review

This study addresses the effect of fruit flies to EtOH with TBI  in lines with and without the expression of the Tau protein. The Tau protein was  reduced by RNAi technology. The TBI’s were induced by concussion of flies in a vial in a standardized approach. The sensitivity to EtOh and mortality were assessed. Flies with a TBI were more sensitive to EtOH exposure within 24 hours and decreased sensitivity to EtOh with TBI after 24 hours. Also, it was demonstrated that flies lacking Tau  were more sensitive to EtOH and TBI with Tau knockdown were in part more sensitive to EtOH.

Minor points:   

1.       Figure 1 legend  : Maybe write ‘MI24’ after ‘mortality rate 24 hours’ just to be clear on Y-Axis labeling.

2.       Might want to use Italic Drosophila throughout the text.

3.       Is it measured anywhere that after TBI  flies are eating or consuming food at the same level as non-TBI flies ?  The desire to eat maybe depressed in TBI flies; thus, increased stress in TBI flies.

The figures and presentation are well presented . No suggestions for modifying the figures or additional experiments.

There are questions a reader may have while reading such a report in flies and related to humans or vertebrates in general.

So, some general comments and thoughts:

1.       Humans drinking more after TBI can be due to so many factors (depression related as  not getting back to normal function). Social pressures flies likely don’t have.

2.       Flies with Drosophila Tau. As mentioned, tau is present in locations in flies but is tau present throughout the CNS of flies  such as in Mushroom bodies and ventral nerve cord (i.e. spinal cord). Does it increase with age  throughout the CNS?

3.       One might want to mention differences in female due to eggs and more fat present to either buffer or add to TBI hits (higher mass) and  EtOH remaining for longer periods of time due to being absorbed into fat.

Future directions-With TBI over days could check  level of Tau in CNS, is it really expressed in CNS  or only in eye and imaginal disks?

Author Response

We are grateful for the thorough reading of our paper and the constructive comments.  Our responses are in blue text below the enumerated comments.  

Minor points:   

  1. Figure 1 legend  : Maybe write ‘MI24’ after ‘mortality rate 24 hours’ just to be clear on Y-Axis labeling.

We agree with the reviewer that this would make the figure legend more clear. We changed the legend in Figure 1 and added MI24 after ‘mortality rate 24 hours’

  1. Might want to use Italic Drosophila throughout the text.

Thank you very much for your comment. We have made sure to italicize all Drosophila throughout the text.

  1. Is it measured anywhere that after TBI  flies are eating or consuming food at the same level as non-TBI flies ?  The desire to eat maybe depressed in TBI flies; thus, increased stress in TBI flies.

We have not measured feeding after TBI, and to our knowledge this has not yet been previously reported.  It is a good point, one which we hope to follow up on in future work. 

The figures and presentation are well presented. No suggestions for modifying the figures or additional experiments.

There are questions a reader may have while reading such a report in flies and related to humans or vertebrates in general.So, some general comments and thoughts:

  1. Humans drinking more after TBI can be due to so many factors (depression related as  not getting back to normal function). Social pressures flies likely don’t have.
  2. Flies with Drosophila Tau. As mentioned, tau is present in locations in flies but is tau present throughout the CNS of flies  such as in Mushroom bodies and ventral nerve cord (i.e. spinal cord). Does it increase with age  throughout the CNS?

In larvae, immunofluorescence studies have shown that dTau proteins are enriched in the fly nervous system including the brain (Br), the ventral nerve chord (VNC) and the peripheral nervous system (PNS) (Burnouf et al. 2016). In the adult Drosophila, immunohistochemistry studies have shown that dTau is expressed in the visual system and central brain, including the mushroom bodies (Papakinolopoulou 2019).  Based on the reviewer’s comment we made this clarification in the introduction of our paper as well.

No age-related studies have looked at the expression of dTau, to our knowledge. However, this is a very interesting point that we would like to pursue.

  1. One might want to mention differences in female due to eggs and more fat present to either buffer or add to TBI hits (higher mass) and  EtOH remaining for longer periods of time due to being absorbed into fat.

We appreciate this comment, as it is a very important and relevant point. We have added the suggested changes in the discussion of the paper.

‘The differences in female ethanol sensitivity could be due to females having eggs and more fat present (higher body mass) to either buffer or add to TBI severity and  EtOH remaining for longer periods due to being absorbed into fat. However, prior studies on humans (PMID: 31789554) indicate that females are more sensitive to the levels of alcohol compared to males. In humans, overall females are more vulnerable than males to many of the health consequences of alcohol use, suggesting that TBI could also play a role in the increased alcohol sensitivity observed in female flies.’

Future directions-With TBI over days could check  level of Tau in CNS, is it really expressed in CNS  or only in eye and imaginal disks?

We appreciate your comment and definitively will consider looking into the expression pattern of Tau in the future, especially as a function of age and gender with and without TBI.

Round 2

Reviewer 1 Report

Comments and Suggestions for Authors

Title: TBI and Tau loss-of-function both affect naïve ethanol sensitivity in Drosophila

The authors answered to some of my concerns, however major concerns are still remained.

1.       Point 8: RT-PCR and/or western blotting must be used to compare the Tau levels of the control vs. TauKO.

The authors answered that “We appreciate your comment. The Drosophila TauKO mutant line was generated in the lab of Burnouf et al. (2016) (reference #47). Using RT-PCR and Western blot analysis, the authors showed that the TauKO line expresses no transcript of the Tau gene, nor does it express a protein. We received this same line from Dr. Skoulakis (reference #85), who replicated the western blot analysis indicating that the line used does not express the Tau protein”

I agree that the previous papers showed the effective of knockdown expression as indicated by ref 47 and 85. Nevertheless, it is widely recognized that transgenic fly cultivation may result in contamination or vector loss; therefore, backcrossing for at least six generations is generally required to ensure that the genotypes are accurate. Since the authors did not conduct backcross, it is difficult to confirm this without employing RT-PCR or WB.

2.       In addition, the consumption recommendation made by reviewer 4 is of the utmost importance. The authors answered that “We have not measured feeding after TBI, and to our knowledge this has not yet been previously reported. It is a good point, one which we hope to follow up on in future work”

I request the authors to perform feeding assay. Accurate interpretation of the whole manuscript cannot be made without this assay.

Comments on the Quality of English Language

-

Reviewer 2 Report

Comments and Suggestions for Authors

Accepted in this version